# A multi-country, multi-year, meta-analytic evaluation of the sex differences in age-specific pertussis incidence rates

**Victoria Peer**®*, **Naama Schwartz, Manfred S. Green**

School of Public Health, University of Haifa, Haifa, Israel

* victoriapapi@yahoo.com

## Abstract

### Background

Pertussis is frequently reported to be more common in females than in males. However, the variability of the sources of these observations makes it difficult to estimate the magnitude and consistency of the sex differences by age. To address this question, we used meta-analytic methods to analyze pertussis national incidence rates by sex and age group from nine countries between the years 1990 and 2017.

### Methods

For each age group, we used meta-analytic methods to combine the female to male incidence rate ratios (RRs) by country and year. Meta-regression was performed to assess the relative contributions of age, country and time-period to the variation in the incidence RRs.

### Results

The pooled female to male incidence RRs (with 95% CI) for ages 0–1, 1–4, 5–9 and 10–14, were 1.03 (1.01–1.06), 1.16 (1.14–1.17), 1.18 (1.15–1.22), 1.15 (1.11–1.18) respectively. For the ages 15–44, 45–64 and 65+ they were 1.65 (1.58–1.72), 1.59 (1.53–1.66), 1.20 (1.16–1.24), respectively. While there were some differences between the countries, the directions were consistent. When including age, country and time in meta-regression analyses, almost all the variation could be attributed to the differences between the age groups.

### Conclusions

The consistency of the excess pertussis incidence rates in females, particularly in infants and very young children, is unlikely to be due to differences in exposure. Other factors that impact on the immune system, including chromosomal differences and hormones, should be further investigated to explain these sex differences. Future studies should consider sex for better understanding the mechanisms affecting disease incidence, with possible implications for management and vaccine development.

**Data Availability Statement:** All relevant data are within the paper and its Supporting Information files. Table number 1 contains all the minimal data used for the study (for meta-analysis). On this table we performed all the data ("minimal data set" for

all reported years together) about pertussis cases by sex (cumulative total n), total population by sex (cumulative total N), calculated incidence rate per 100 000 male or female population, and incidence rate ratio (female: male incidence rate ratio). All this data required for study findings replication. All raw data by single year, age group and sex for each country (pertussis cases by sex n by single year and total population by sex N by single year) are available in the Supporting Information.

**Funding:** This research did not receive any specific grant from funding agencies in the public, commercial, or not-for-profit sectors.

**Competing interests:** We declare no conflicts of interest.

## Introduction

Despite the availability of an effective vaccine, pertussis caused by ***Bordetella pertussis*** (B. pertussis) remains a public health problem in both developing and developed countries [1]. Clinical manifestations of the disease can be mild, severe with occasional fatal outcomes, especially in infants [2]. The appearance of new cases could be, for example, due to low immunization rates [3] inadequate immune responses to vaccine [4] or waning immunity following immunization [5].

Reports from individual countries often mention higher pertussis incidence rates (IR) in females [6–11], not always specifying age groups. These observations have usually based on data from individual countries or using case series data, without denominators for calculating incidence rates (IR). If the excess in females is consistent, it could be due to a number of factors.

These include response to vaccine, behavioral and social factors [12], chromosomal factors, or an interaction between sex hormones and immune function affecting the clinical manifestations of the disease [13]. Establishment of the magnitude and consistency of the sex differences in the disease can provide important clues to the mechanism of disease. In order to study this further, we carried out an in-depth study of the sex differences in pertussis incidence rates at different ages, in different countries and over a number of years, based on national data.

## Materials and methods

### Source of data and search strategy

In order to guarantee the data quality, we restricted our search strategy to all countries in Europe, North and South America, Australia and New Zealand, with established diagnostic tools and well-organized mandatory reporting systems, that provide data by age and sex for a number of years. National data were obtained either from official internet sites or by contacting representatives of the appropriate country health authorities. The original search was performed from March to June of 2018.

There were nine countries for which the national data were available by age, sex and year—Australia (for years 2001–2016), Canada (for years 1991–2015), Czech Republic (for years 2008–2013), England (for years 1990–2016), Finland (for years 1995–2016), Israel (for years 1998–2016), Netherlands (for years 2001–2017), New Zealand (for years 1997–2015), and Spain (for years 2005–2015).

Data for Australia were obtained from the National Notifiable Diseases Surveillance System (NNDSS), the Department of Health [14], for Canada from the Canadian Notifiable Disease Surveillance System (CNDSS) [15], for the Czech Republic from the Institute of Health Information and Statistics [16], for England, directly from Public Health England (PHE), for Finland, from the National Institute for Health and Welfare (THL) [17], for Israel, from the Ministry of Health, for the Netherlands, directly from the official representative of RIVM, for New Zealand, from the Institute of Environmental Science and Research (ESR) for the Ministry of Health [18], and for Spain, from the Spanish Epidemiological Surveillance Network at the National Centre for Epidemiology [19]. Data on the population size by age, sex and year for the Australian population was obtained from the Australian Bureau of Statistics [20] and for Canada from Statistics Canada [21], for the Czech Republic from the Czech Statistical Office [22], for England, from the Population Estimates Unit, Population Statistics Division, Office for National Statistics [23], and for Finland from the Statistics Finland's PX-Web databases [24]. Data for Israel were obtained from the Central Bureau of Statistics [25], for

Netherlands from Statistics Netherlands' database [26], for New Zealand from Stats NZ, Info-share, New Zealand [27], and for Spain from the Demographic Statistics Database [28].

### Ethical considerations and informed consent

National, open access aggregative and anonymous data were used and there was no need for ethics committee approval.

### Statistical analyses

The period under study was between 1990 and 2017. Due to the large amount of data, for presentation purposes, the years were grouped for the graphical presentations. Annual pertussis incidence rates (per 100,000) were calculated by sex and age group, for each country and group of years using the number of reported cases divided by the respective population size and multiplied by 100,000.

The age groups considered were <1 (infants), 1–4 (early childhood), 5–9 (late childhood), 10–14 (puberty), 15–44 or 15–39 (young adulthood), 45–64 or 40–59 (middle adulthood) and 65+/60+ (senior adulthood) years. The surveillance systems in Canada, England, Finland, Netherlands, and New Zealand used similar age groups except for the following: 15–39, 40–59 and 60+. For Australia and Finland, data are missing for ages <1 and 1–4 separately. We made an informed decision not to combine these age groups since there is a difference between infants <1 year old and early childhood. The female to male incidence rate ratio (RR) was calculated by dividing the annual incidence rate in females by that of males, by age group, country, and time periods.

The data were analyzed using meta-analytic methods and meta-regression STATA software version 12.1 (Stata Corp., College Station, TX). For the purpose of applying meta-analytic methods, the national data sets for each age group by country and year were considered as separate "studies" and the outcome variable was the female to male incidence RR. After obtaining pooled incidence RRs separately for each age group, by country and time period, pooled incidence RR's for each age group were obtained for all countries and time periods together. The results are presented in forest plots. Heterogeneity was evaluated using Cochran's Q statistic, and $Tau^2$ and $I^2$ (to estimate between-study variance) [29]. Where significant heterogeneity was present (if $I^2 \geq 50\%$ and/or the Q test yielded a p-value <0.1) the random effects model [30] was used to estimate pooled RRs and 95% confidence intervals (CI). Otherwise, the fixed effects model was used.

We performed leave-one-out sensitivity analysis in order to determine how each country and group of years affected the outcome following the recomputed pooled pertussis female to male incidence RR. In order to determine whether there were countries or time periods that are outliers, we created funnel plots and used Egger's test. In order to explore the contributions of age, countries and time periods to the heterogeneity of the incidence RRs, meta-regression analyses were performed, with incidence RR as the dependent variable.

## Results

The summary of male and female pertussis incidence rates (per 100,000 populations) in different countries for each age group and for number of years is presented in Table 1.

Age-specific rates by sex were highest in female infants, in 1–4-year-olds and in age groups of 5–9 and 10–14. There was a decrease in the incidence rate of pertussis in adults, in both groups of males and females.

**Table 1. Details of the countries included in the meta-analysis, by sex and age group—Descriptive data.**

| | | | Females | | Males | | |
| --- | --- | --- | --- | --- | --- | --- | --- |
| Age | Country | Years | n/N | IR | n/N | IR | RR |
| <1 | Canada | 1991–2015 | 5113/4446799 | 115 | 5297/4682619 | 113.1 | 1.02 |
| | Czech Republic | 2008–2013 | 76/332712 | 22.8 | 83/349195 | 23.8 | 0.96 |
| | England | 1990–2016 | 4829/8306732 | 58.1 | 4651/8725051 | 53.3 | 1.09 |
| | Israel | 1998–2016 | 1397/1410400 | 99 | 1587/1486100 | 106.8 | 0.93 |
| | Netherlands | 2001–2017 | 1393/1540059 | 90.5 | 1378/1616870 | 85.2 | 1.06 |
| | New Zealand | 1997–2015 | 1309/548520 | 238.6 | 1283/576900 | 222.4 | 1.07 |
| | Spain | 2005–2015 | 3169/2514548 | 126 | 3318/2679186 | 123.8 | 1.02 |
| 1–4 | Canada | 1991–2015 | 11781/18225737 | 64.6 | 10770/19156418 | 56.2 | 1.15 |
| | Czech Republic | 2008–2013 | 95/1343670 | 7.1 | 77/1410748 | 5.5 | 1.30 |
| | England | 1990–2016 | 10149/33207057 | 30.6 | 9066/34821935 | 26 | 1.17 |
| | Israel | 1998–2016 | 1404/5443300 | 25.8 | 1229/5731500 | 21.4 | 1.20 |
| | Netherlands | 2001–2017 | 3081/6327474 | 48.7 | 2888/6632134 | 43.5 | 1.12 |
| | New Zealand | 1997–2015 | 2397/2191980 | 109.4 | 2333/2308880 | 101 | 1.08 |
| | Spain | 2005–2015 | 2309/10233932 | 22.6 | 1996/10880587 | 18.3 | 1.23 |
| 5–9 | Australia | 2001–2016 | 19018/10814642 | 175.9 | 17996/11398585 | 157.9 | 1.11 |
| | Canada | 1991–2015 | 13502/23469919 | 57.5 | 11925/24668602 | 48.3 | 1.19 |
| | Czech Republic | 2008–2013 | 204/1450621 | 14.1 | 154/1532669 | 10 | 1.40 |
| | England | 1990–2016 | 9009/41012194 | 22 | 7490/42989082 | 17.4 | 1.26 |
| | Finland | 1995–2016 | 1332/3297629 | 40.4 | 1109/3440956 | 32.2 | 1.25 |
| | Israel | 1998–2016 | 2742/6287700 | 43.6 | 2583/6616300 | 39 | 1.12 |
| | Netherlands | 2001–2017 | 6953/8108728 | 85.7 | 6395/8494005 | 75.3 | 1.14 |
| | New Zealand | 1997–2015 | 2823/2752910 | 102.5 | 2528/2899540 | 87.2 | 1.18 |
| | Spain | 2005–2015 | 2193/12287011 | 17.8 | 1899/13017097 | 14.6 | 1.22 |
| 10–14 | Australia | 2001–2016 | 18054/10797396 | 167.2 | 17501/11377822 | 153.8 | 1.09 |
| | Canada | 1991–2015 | 9632/24391864 | 39.5 | 8662/25685783 | 33.7 | 1.17 |
| | Czech Republic | 2008–2013 | 829/1339518 | 61.9 | 733/1416001 | 51.8 | 1.20 |
| | England | 1990–2016 | 3048/40624659 | 7.5 | 2787/42597565 | 6.5 | 1.15 |
| | Finland | 1995–2016 | 1678/3375446 | 49.7 | 1294/3522497 | 36.7 | 1.35 |
| | Israel | 1998–2016 | 2659/5807300 | 45.8 | 2624/6106400 | 43 | 1.07 |
| | Netherlands | 2001–2017 | 10679/8277833 | 129 | 10023/8668277 | 115.6 | 1.12 |
| | New Zealand | 1997–2015 | 1773/2776650 | 63.9 | 1740/2919850 | 59.6 | 1.07 |
| | Spain | 2005–2015 | 1968/11627137 | 16.9 | 1686/12301238 | 13.7 | 1.23 |
| 15-39/15-44 | Australia | 2001–2016 | 35706/72741755 | 49.1 | 23400/75391102 | 31.8 | 1.54 |
| | Canada | 1991–2015 | 7924/140453550 | 5.6 | 4508/143987472 | 3.1 | 1.80 |
| | Czech Republic | 2008–2013 | 1105/12978912 | 8.5 | 856/13725818 | 6.2 | 1.37 |
| | England | 1990–2016 | 5222/233399206 | 2.2 | 3433/234901126 | 1.5 | 1.53 |
| | Finland | 1995–2016 | 2036/18050351 | 11.3 | 1285/18898064 | 6.8 | 1.66 |
| | Israel | 1998–2016 | 5073/29264100 | 17.3 | 3014/29586200 | 10.2 | 1.70 |
| | Netherlands | 2001–2017 | 14916/44687694 | 33.4 | 10414/45667655 | 22.8 | 1.46 |
| | New Zealand | 1997–2015 | 4912/13976900 | 35.1 | 2586/13546700 | 19.1 | 1.84 |
| | Spain | 2005–2015 | 2254/105413400 | 2.1 | 1195/110542308 | 1.1 | 1.98 |
| 40-59/45-64 | Australia | 2001–2016 | 37571/42573071 | 88.3 | 24409/41988401 | 58.1 | 1.52 |
| | Canada | 1991–2015 | 3037/109655649 | 2.8 | 1956/110461323 | 1.8 | 1.56 |
| | Czech Republic | 2008–2013 | 242/8624880 | 2.8 | 107/8403729 | 1.3 | 2.20 |
| | England | 1990–2016 | 4433/177644620 | 2.5 | 3030/175100277 | 1.7 | 1.44 |
| | Finland | 1995–2016 | 1364/16307550 | 8.4 | 553/16513241 | 3.3 | 2.50 |
| | Israel | 1998–2016 | 2159/13327000 | 16.2 | 1377/12368500 | 11.1 | 1.46 |
| | Netherlands | 2001–2017 | 11087/40278130 | 27.5 | 8018/40902904 | 19.6 | 1.40 |
| | New Zealand | 1997–2015 | 3697/10685350 | 34.6 | 2349/10201030 | 23 | 1.50 |
| | Spain | 2005–2015 | 1030/64340310 | 1.6 | 524/63103755 | 0.8 | 1.93 |

*(Continued)*

**Table 1.** (Continued)

| Age | Country | Years | Females | | Males | | |
|---|---|---|---|---|---|---|---|
| | | | n/N | IR | n/N | IR | RR |
| 60+/65+ | Australia | 2001–2016 | 23102/25538457 | 90.5 | 17163/21417772 | 80.1 | 1.13 |
| | Canada | 1991–2015 | 687/78346403 | 0.9 | 477/64590224 | 0.7 | 1.19 |
| | Czech Republic | 2008–2013 | 83/5999018 | 1.4 | 35/4087584 | 0.9 | 1.62 |
| | England | 1990–2016 | 1389/163257756 | 0.9 | 1040/129663953 | 0.8 | 1.06 |
| | Finland | 1995–2016 | 359/15066114 | 2.4 | 182/11159619 | 1.6 | 1.46 |
| | Israel | 1998–2016 | 858/7566600 | 11.3 | 585/5759300 | 10.2 | 1.12 |
| | Netherlands | 2001–2017 | 6791/32663089 | 20.8 | 4544/27098379 | 16.8 | 1.24 |
| | New Zealand | 1997–2015 | 1618/7386000 | 21.9 | 1130/6302700 | 17.9 | 1.22 |
| | Spain | 2005–2015 | 340/49879431 | 0.7 | 182/37127234 | 0.5 | 1.39 |

IR = incidence rate, IR per 100 000 Male or Female population, incidence RR = female: male incidence Rate Ratio

n- Cumulative total of pertussis cases for given years.

N- Cumulative total of the population for given years.

Infants = age<1 year; early childhood = 1–4 years; late childhood = 5–9 years; puberty = 10–14 years; young adulthood = 15–44 or 15–39 years; middle adulthood = 40–59 or 45–64 years; senior adulthood = 60+ or 65+ years.

Results of the study are presented in the forest plots presented by age group in Figs 1–7 (with CI = 95% confidence interval, RR = rate ratio. The right side of the X-axis indicates a higher IR for females and the left side for males).

The forest plot for infants (age <1) is shown in Fig 1.

The forest plots for age 15-44/15-39 and 45-64/40-59 are shown in Figs 5 and 6 respectively. For age 15-44/15-39, Fig 5, the overall incidence RR = 1.65 (95% CI 1.58–1.72), with $I^2$ = 90.7%, and $Tau^2$ = 0.0135. Female dominance is significant for every country population, ranging from an incidence RR = 1.33 for Czech Republic to 1.96 for Spain RR = 1.96.

To evaluate the effect of individual country and the group of years on the pooled RR, we performed leave-one-out sensitivity analysis and recomputed the pooled RRs. After omitting one country at a time, the pooled RRs remained very similar (Table 2).

For the funnel plot (Fig 8), Egger's test p value for asymmetry was not significant for all age groups except middle adulthood (from infancy, young childhood, late childhood, puberty, young adulthood, and senior adulthood p value were p = 0.711, p = 0.427, p = 0.217, p = 0.176, p = 0.055 and p = 0.076 respectively). Evidence of asymmetry was observed only for middle adulthood with p = 0.036.

## Discussion

In this study, we examined the sex differences in pertussis incidence rates by age group in nine countries over a period of six to 27 years. These results revealed higher pertussis incidence rates in females than in males in all age groups from infancy to older adults. The pooled results varied by age from a 3% excess in infants to an excess of 65% in in young adulthood. These findings were consistent over countries and time periods. The meta-regression results revealed that among the variables, age group contributed almost all the variation in the incidence RRs. The results of this study contrast with the perception that males suffer more than females from infectious diseases [13, 31].

Surveillance data from 1995 onwards in former West German states showed a higher pertussis incidence in females (overall 60% of cases) than in males mainly due to a higher proportion of females among adult cases [4]. In England, slight differences were observed between

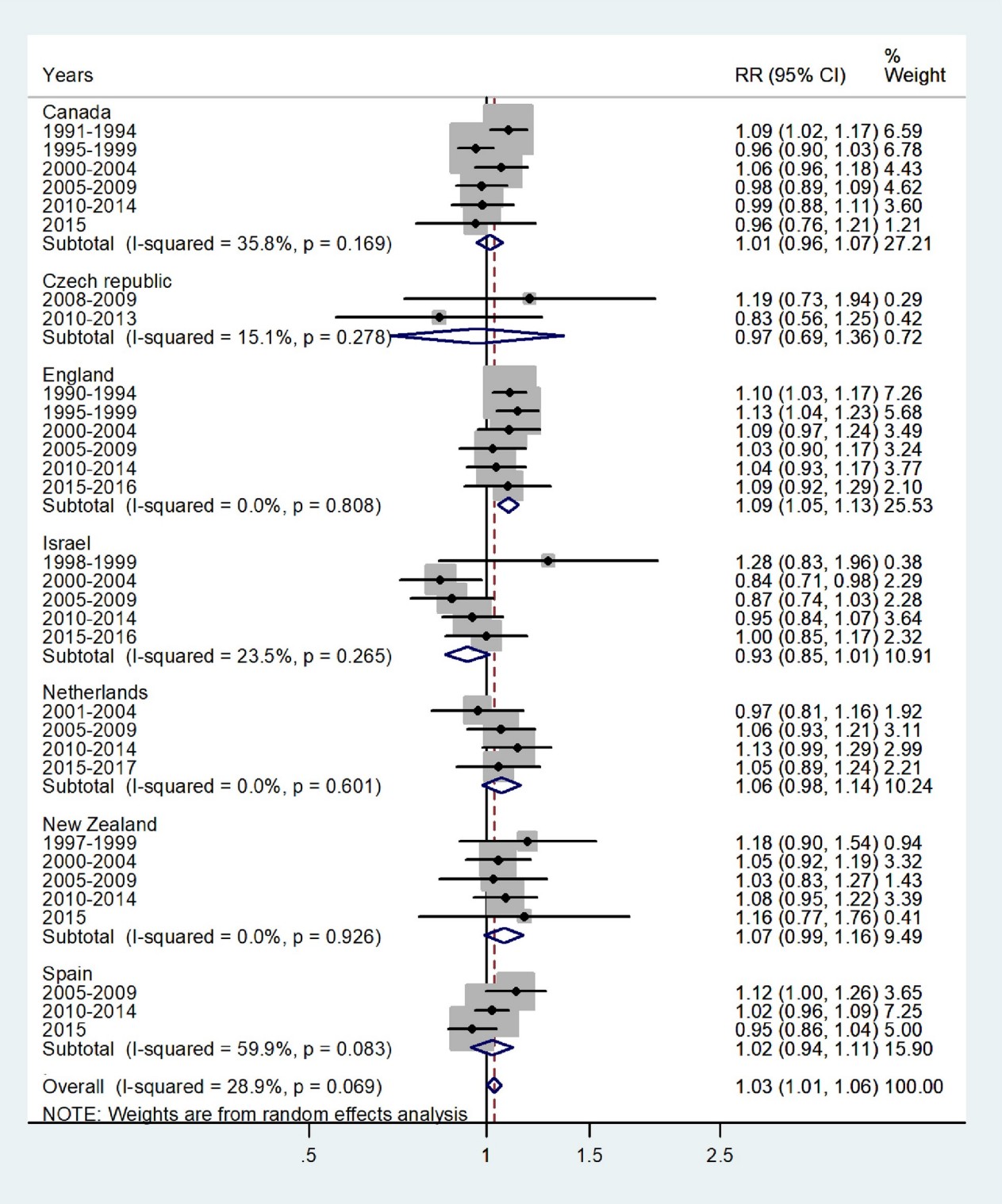

**Fig 1. Forest plot of the female to male pertussis incidence rate ratios (RR) for different years in Canada, Czech Republic, England, Israel, Netherlands, New Zealand, and Spain in infants.** The overall incidence RR in infants was 1.03 (95% CI 1.01–1.06), which indicated a small, but significant increase in incidence of disease in female infants, with low heterogeneity, $I^2$ = 28.9%, and $Tau^2$ = 0.002. The incidence RR in infancy varied from 0.93 in Israel to 1.09 in England. The forest plot for ages 1–4 is shown in Fig 2.

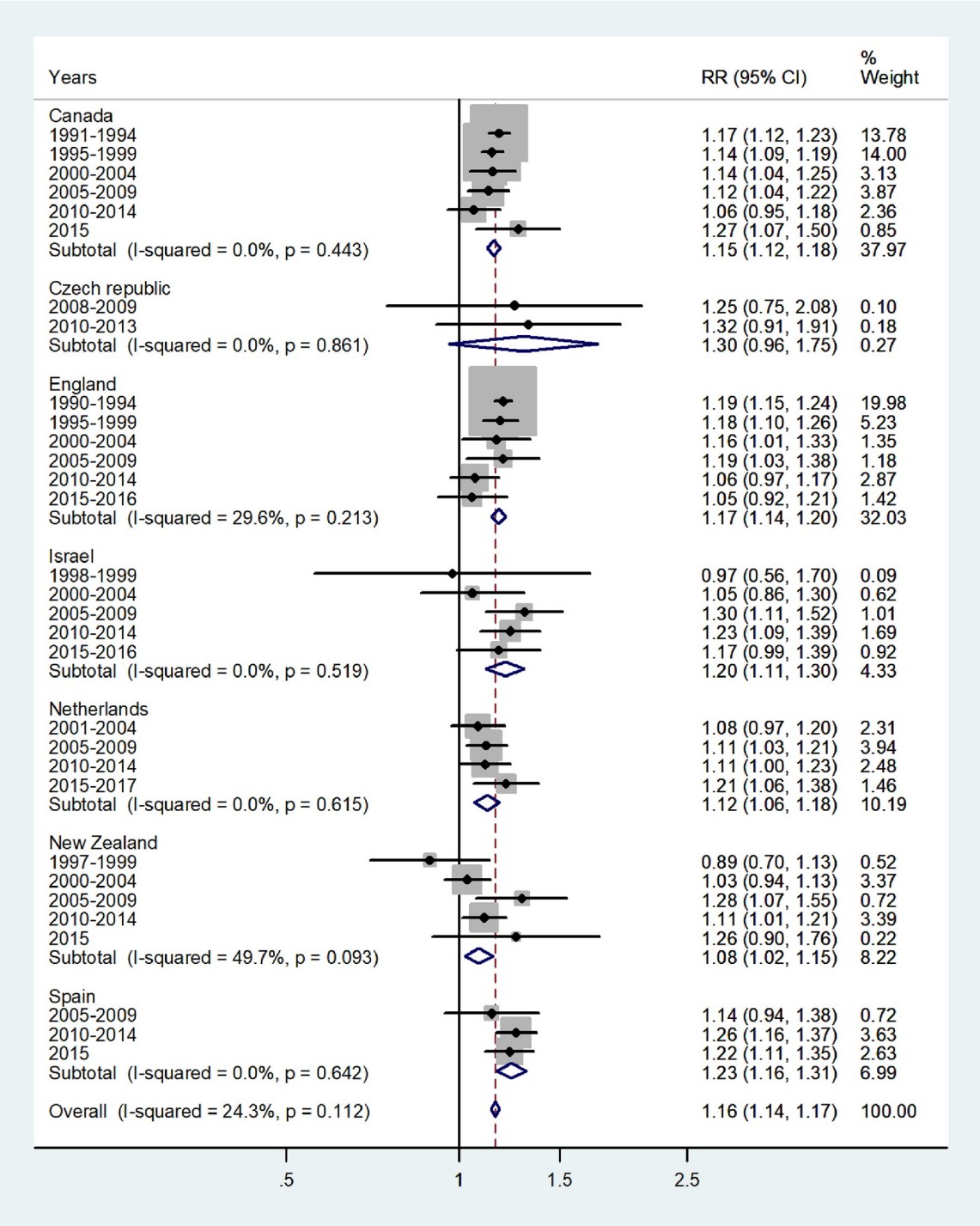

**Fig 2. Forest plot of the female to male pertussis incidence rate ratios (RR) for different years in Canada, Czech Republic, England, Israel, Netherlands, New Zealand, and Spain in yearly childhood.** The overall incidence RR in ages 1–4 was 1.16 (95% CI 1.14–1.17), which indicated a 16% excess incidence rates in females, with low heterogeneity, $I^2$ = 24.3%, and $Tau^2$ = 0.001. The subtotal incidence RRs varied from 1.08 in New Zealand to 1.30 in Czech Republic. The forest plot for age 5–9 is shown in Fig 3.

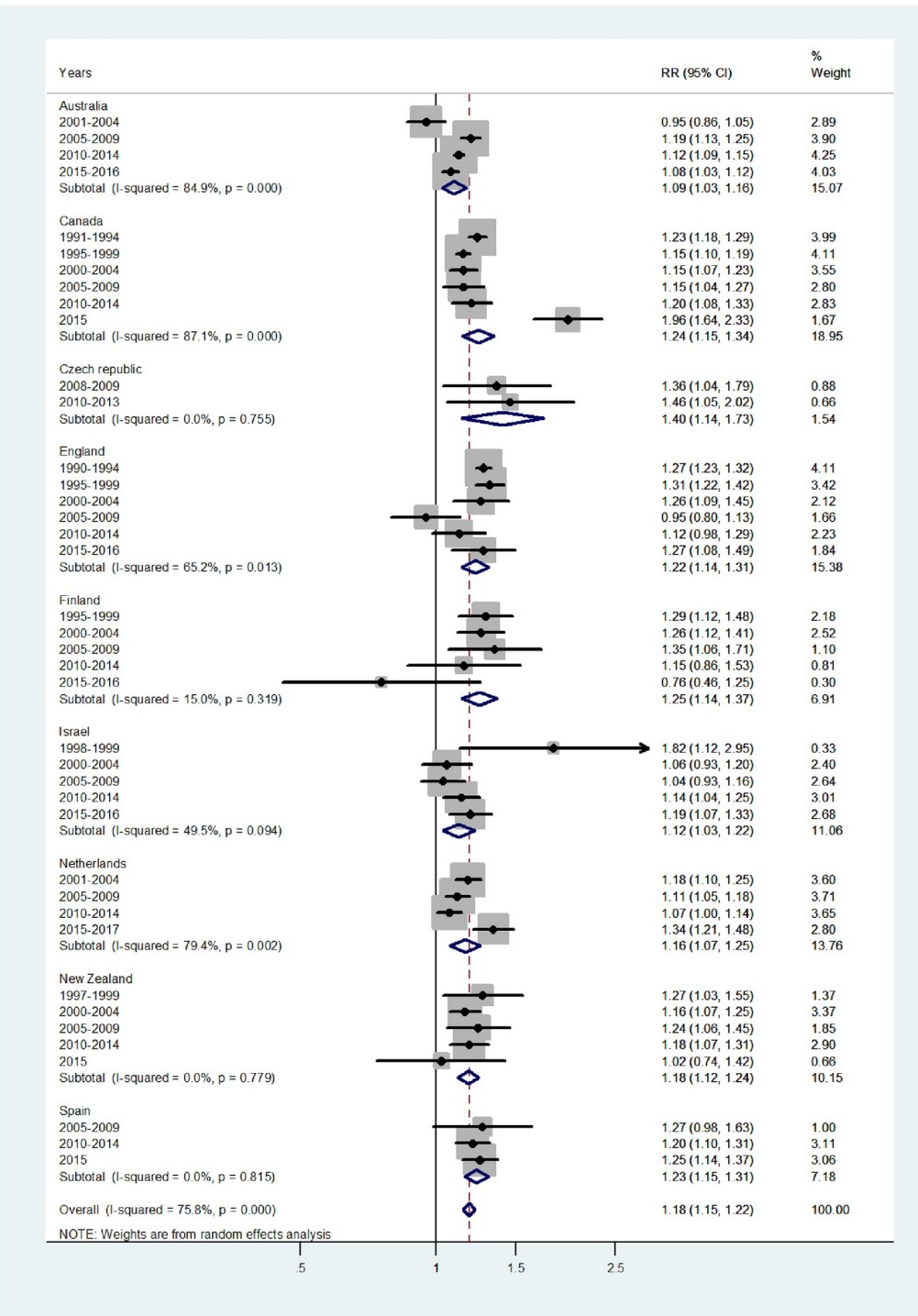

**Fig 3. Forest plot of the female to male pertussis incidence rate ratios (RR) for different years in Australia, Canada, Czech Republic, England, Finland, Israel, Netherlands, New Zealand, and Spain in late childhood.** The overall incidence RR for age 5–9 was 1.18 (95% CI 1.15–1.22), which indicated 18% excess incidence rates in females, with $I^2$ = 75.8%, and $Tau^2$ = 0.005. The subtotal incidence RRs are significantly greater than 1 in all countries and varied from 1.09 in Australia to 1.4 in Czech Republic. The forest plot for age 10–14 is shown in Fig 4.

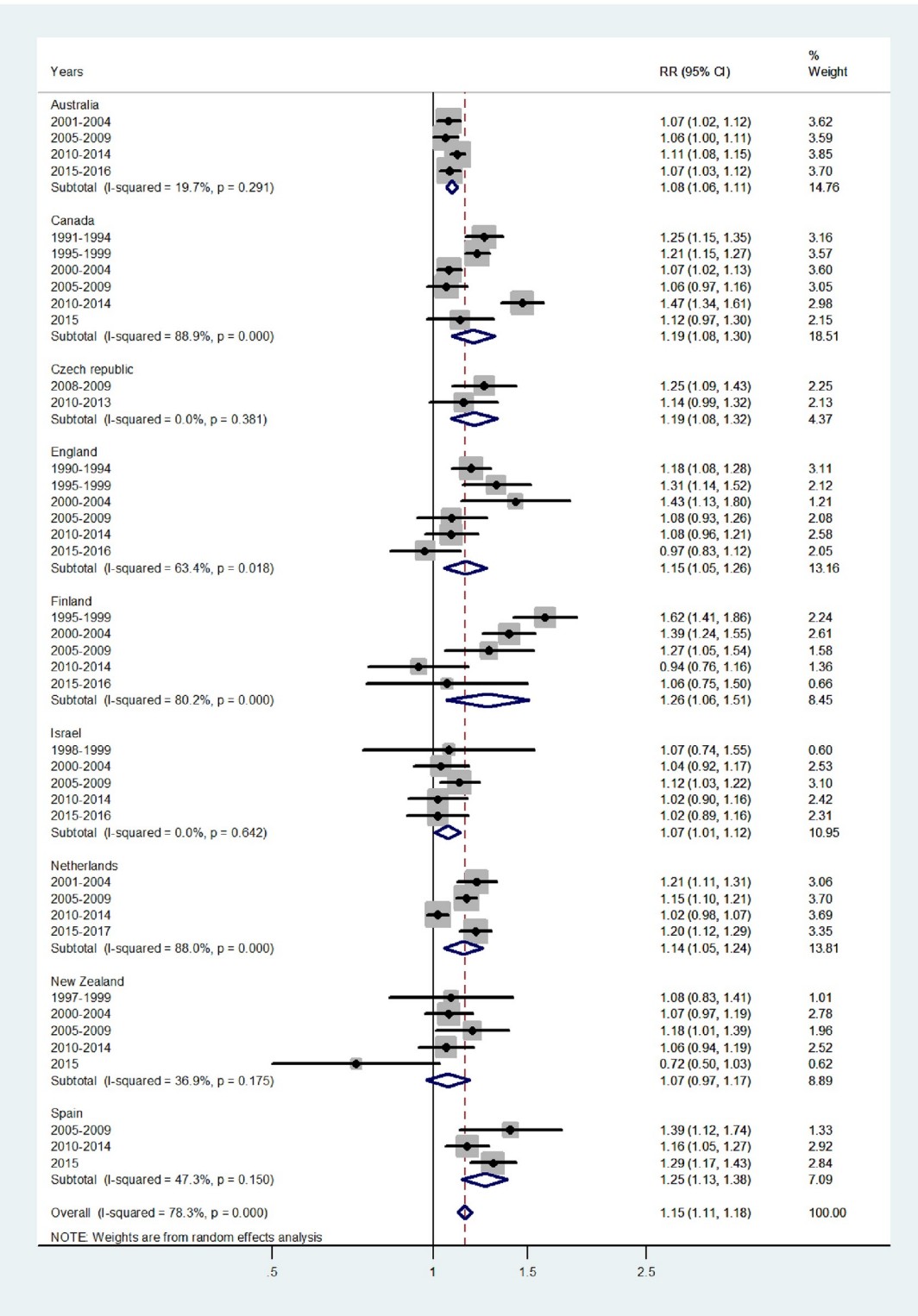

**Fig 4. Forest plot of the female to male pertussis incidence rate ratios (RR) for different years in Australia, Canada, Czech Republic, England, Finland, Israel, Netherlands, New Zealand, and Spain in puberty.** The overall incidence RR at age 10–14 was 1.15 (95% CI 1.11–1.18), with $I^2$ = 78.3%, and $Tau^2$ = 0.0062. The subtotal incidence RRs varied from 1.07 in Israel and New Zealand to 1.26 in Finland.

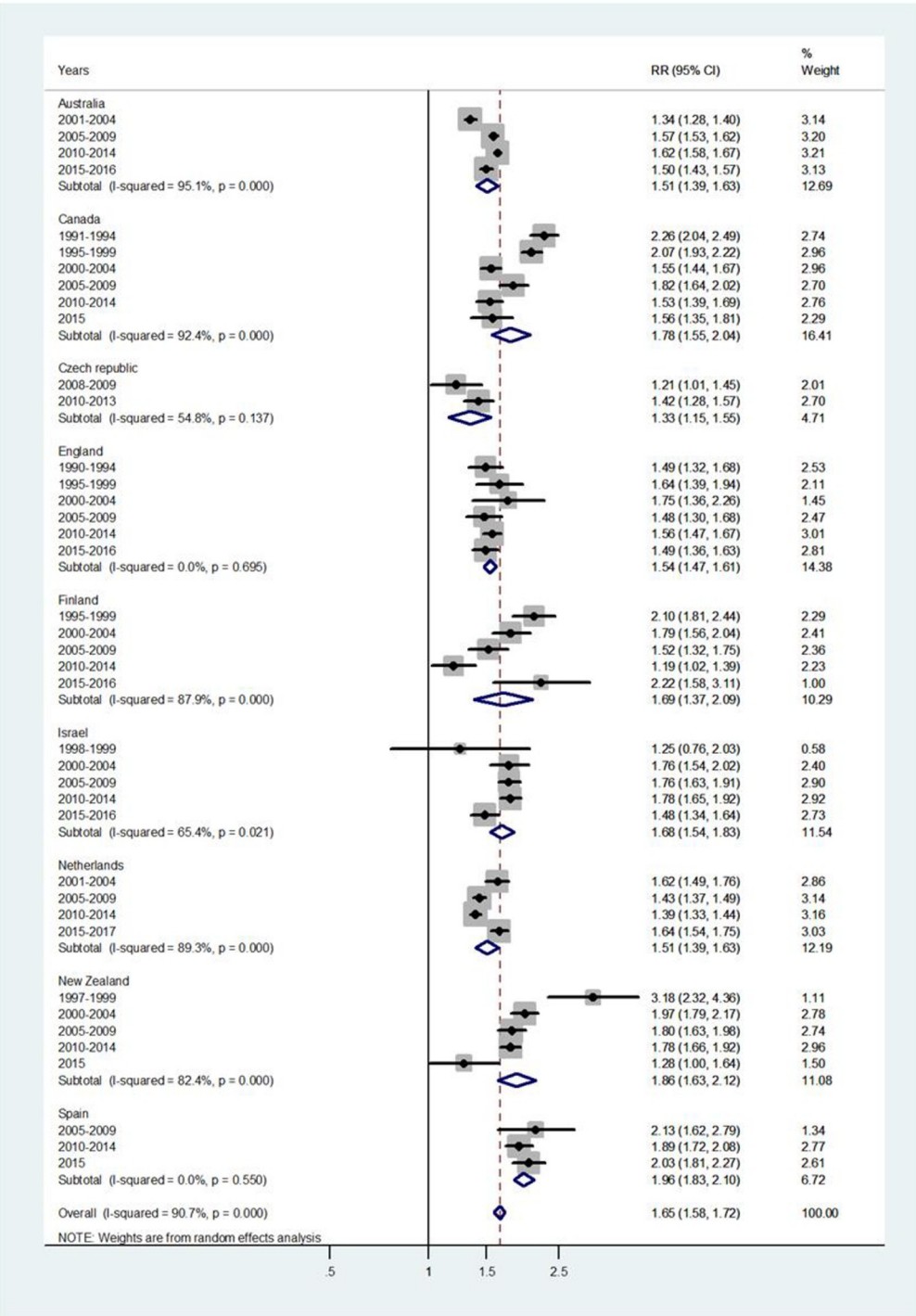

**Fig 5. Forest plot of the female to male pertussis incidence rate ratios (RR) for different years in Australia, Canada, Czech Republic, England, Finland, Israel, Netherlands, New Zealand, and Spain in young adulthood.** For age 45-64/ 40-59 (Fig 6), the overall incidence RR = 1.59, 95% CI 1.53–1.66, $I^2$ = 85.9%, and $Tau^2$ = 0.0106, ranging from an incidence RR = 1.4 in Netherlands to an RR = 2.43 in Finland.

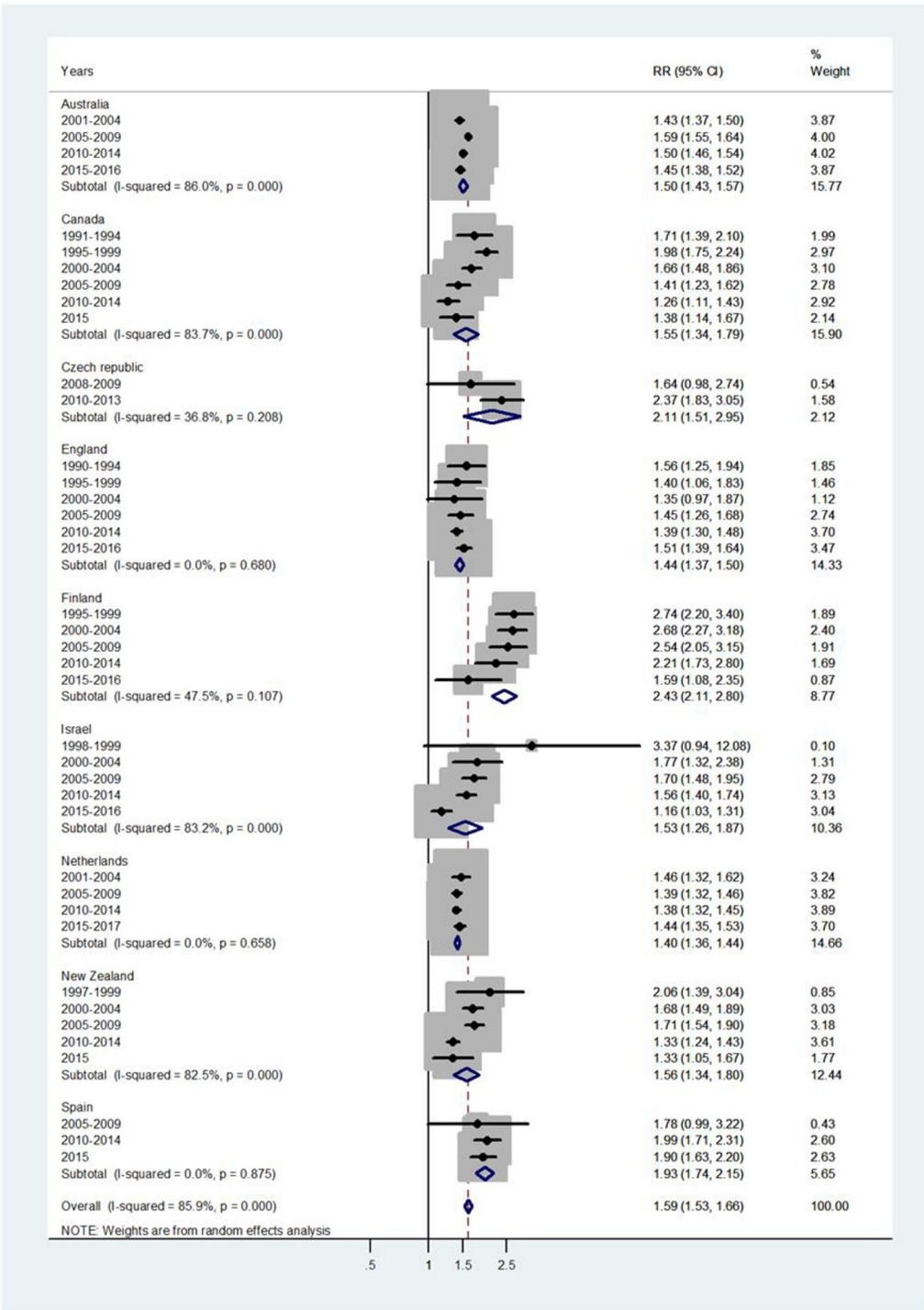

**Fig 6. Forest plot of the female to male pertussis incidence rate ratios (RR) for different years in Australia, Canada, Czech Republic, England, Finland, Israel, Netherlands, New Zealand, and Spain in middle adulthood.** At age 60+/65 +, the overall RR = 1.2, 95% CI 1.16–1.24, $I^2$ = 51.2%, and $Tau^2$ = 0.0034, ranging from RR = 1.12 in England to RR = 1.53 in Czech Republic. The forest plot at age 60+/65+ is presented in Fig 7.

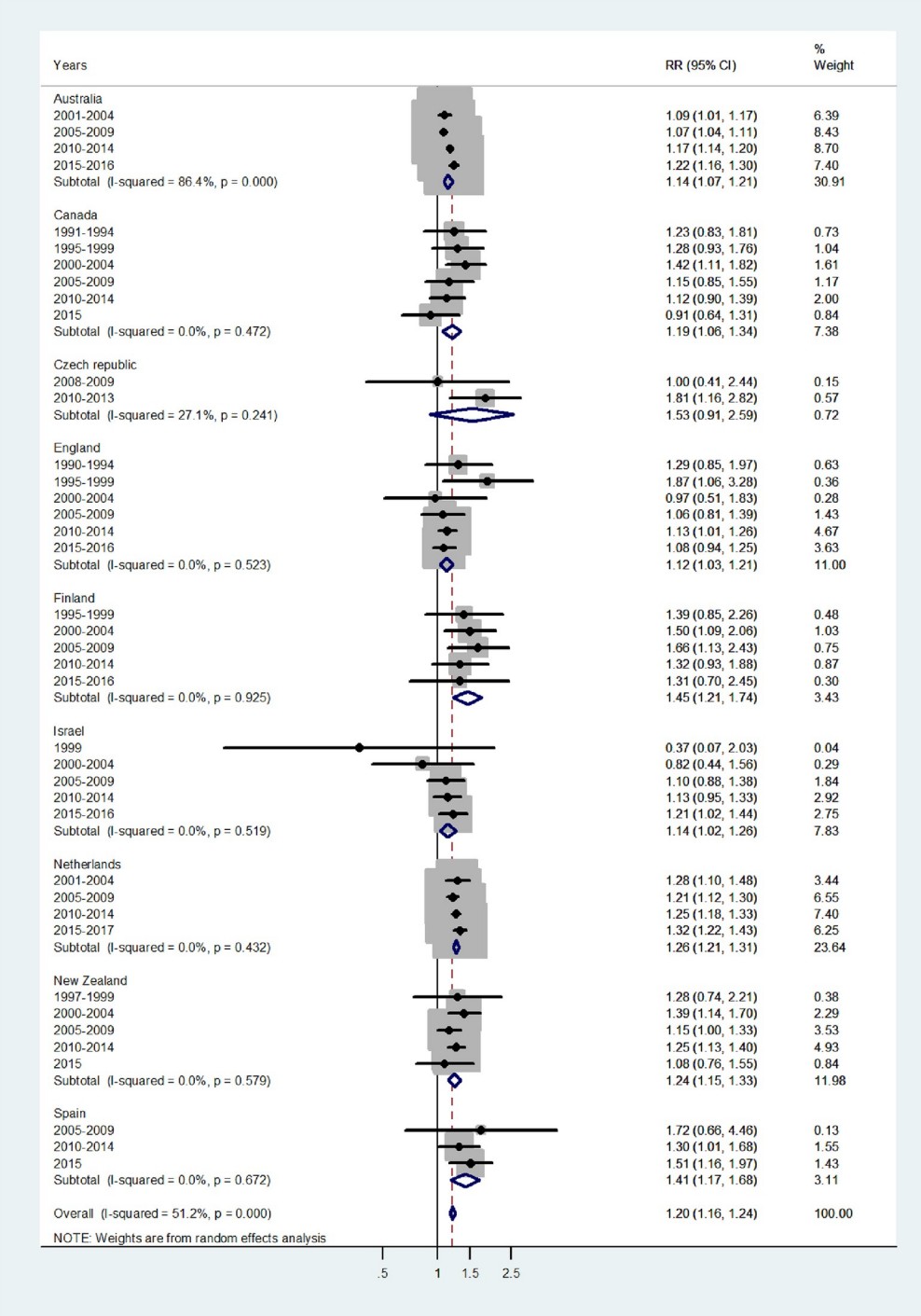

**Fig 7. Forest plot of the female to male pertussis incidence rate ratios (RR) for different years in Australia, Canada, Czech Republic, England, Finland, Israel, Netherlands, New Zealand, and Spain in senior adulthood.**

males and females, with 48.1% incidence rate in male and 51.9% in female during 2011–2012, including in those aged 10–19 years [8]. Skoff et al [9] revealed age specific transmission of pertussis over time and observed that in United States, during 2000–2016, among all pertussis cases reported, the majority (54.7%) were female. In a study in Barcelona, similar incidences of

**Table 2. Sensitivity analysis, by age group and country.**

| Country Removed | Infants RR (CI) | Early Childhood RR (CI) | Late Childhood RR (CI) | Puberty RR (CI) | Young Adulthood RR (CI) | Middle Adulthood RR (CI) | Senior Adulthood RR (CI) |
|---|---|---|---|---|---|---|---|
| | | | | | Age Group | | |
| Australia | - | - | 1.2 (1.15–1.24) | 1.16 (1.11–1.21) | 1.66 (1.53–1.8) | 1.68 (1.52–1.86) | 1.22 (1.14–1.3) |
| Canada | 1.03 (0.98–1.09) | 1.16 (1.14–1.19) | 1.19 (1.13–1.24) | 1.15 (1.1–1.2) | 1.62 (1.53–1.73) | 1.65 (1.53–1.79) | 1.2 (1.13–1.27) |
| Czech Republic | 1.03 (0.99–1.08) | 1.16 (1.14–1.18) | 1.18 (1.14–1.22) | 1.15 (1.1–1.19) | 1.68 (1.57–1.79) | 1.61 (1.5–1.73) | 1.19 (1.13–1.26) |
| England | 1.02 (0.98–1.06) | 1.15 (1.13–1.17) | 1.17 (1.13–1.21) | 1.15 (1.1–1.2) | 1.66 (1.55–1.78) | 1.67 (1.55–1.81) | 1.22 (1.15–1.3) |
| Finland | - | - | 1.18 (1.13–1.22) | 1.13 (1.09–1.17) | 1.64 (1.54–1.76) | 1.54 (1.47–1.61) | 1.18 (1.12–1.24) |
| Israel | 1.05 (1.02–1.08) | 1.15 (1.14–1.17) | 1.19 (1.15–1.24) | 1.16 (1.11–1.21) | 1.64 (1.53–1.75) | 1.67 (1.54–1.8) | 1.21 (1.14–1.29) |
| Netherlands | 1.02 (0.98–1.07) | 1.16 (1.14–1.18) | 1.19 (1.14–1.25) | 1.16 (1.1–1.21) | 1.67 (1.56–1.79) | 1.69 (1.55–1.83) | 1.19 (1.12–1.26) |
| New Zealand | 1.02 (0.98–1.07) | 1.16 (1.14–1.18) | 1.19 (1.14–1.24) | 1.16 (1.11–1.21) | 1.62 (1.52–1.72) | 1.66 (1.54–1.8) | 1.2 (1.12–1.27) |
| Spain | 1.03 (0.98–1.09) | 1.15 (1.13–1.17) | 1.18 (1.13–1.23) | 1.14 (1.1–1.19) | 1.61 (1.52–1.71) | 1.61 (1.5–1.72) | 1.18 (1.12–1.25) |

RR = rate ratio; CI = confidence interval

Similar results were obtained after dropping one group of years at a time (Table 3).

pertussis were observed in males and females under the age 12[10]. In Alberta, Canada, between 2004 and 2015, incidence rates by sex in children under the age 14 were similar between females and males [11].

Unlike in the present study, in general, the sex differences in incidence rates were not reported by age or whether they were consistent over time periods. In some, studies were based on hospital or local data, without population denominators [6–10]. This could be an important source of selection bias.

The current study is based on national data with very large populations, covering a number of years and consequently with large numbers of cases. Selection bias has been minimized by using national data over different time periods, which should be representative of each country. Relevant denominators were available to compute incidence rates as opposed to studies based on a case series. The inclusion of nine countries, with advanced health system, allowed us to evaluate the consistency of the findings over different populations and many years. There is no evidence to suggest that there is selective care or differences in vaccine coverage according to the sex of the child, in any of the countries in this study. Underreporting, as the result of non-specific clinical manifestations of the disease and the lack of laboratory confirmation, may be a source of information bias [32]. However, this is unlikely to be different for females and males. There may be a difference in use of health services by sex in the adult age groups [33], but is unlikely to be a factor in infants and children in the countries in this study. Surveillance systems as well as the diagnostic criteria and proportion of laboratory-confirmed cases are heterogeneous [34], but should not differ between females and males.

There is no clear evidence on differences in response to pertussis vaccine between males and females. Antibody levels have been found to be similar in males and females in infants, children and adults following immunization [35, 36]. As regards exposure differences, in young and middle adulthood, women may have more exposure to cases of pertussis while

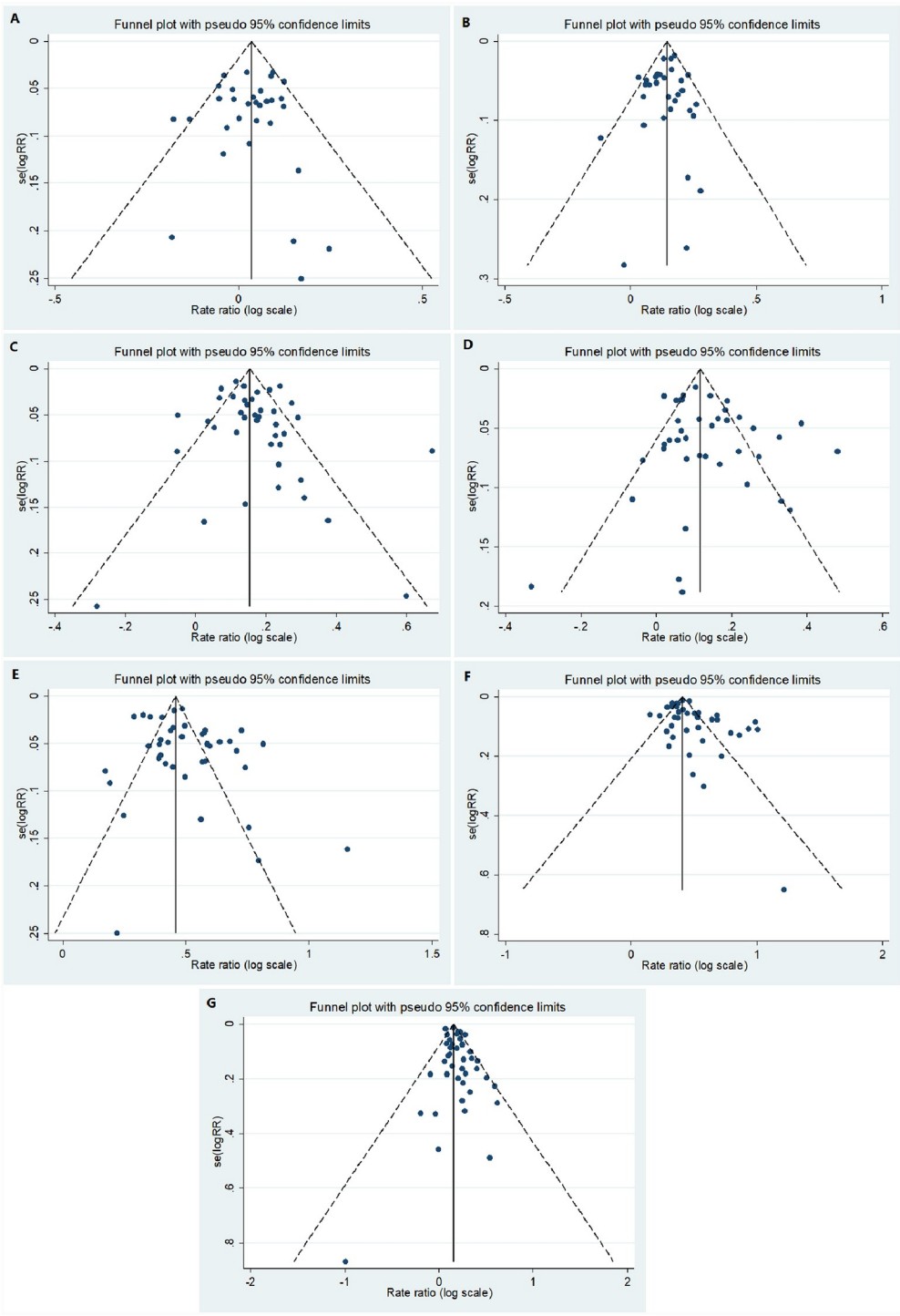

**Fig 8. Funnel plots: A) for infants, B) for early childhood, C) late childhood, D) for puberty, E) for young adulthood, F) for middle adulthood and G) for senior adulthood.** In the meta-regression analyses, including age group, country and year, age group contributed almost all the variation in the incidence RRs. For infants the incidence RR was lower than for other age groups and in young and middle adulthood, the incidence RRs were significantly higher than in the other age groups (P < .0001). There was no significant association with reporting time-periods, aside from a borderline negative trend among puberty and middle adulthood groups. In those groups, as the time-periods increased, the incidence RR values decreased (P = 0.05 and P = 0.05 for puberty and middle adulthood respectively). In this group, as the time-periods increased, the incidence RR values decreased, (P = 0.02).

**Table 3. Sensitivity analysis, by age group and years.**

| Years Removed | Infants RR (CI) | Early Childhood RR (CI) | Late Childhood RR (CI) | Puberty RR (CI) | Young Adulthood RR (CI) | Middle Adulthood RR (CI) | Senior Adulthood RR (CI) |
|---|---|---|---|---|---|---|---|
| | | | | Age Group | | | |
| **1990–1994** | 1.02 (0.997–1.04) | 1.14 (1.12–1.16) | 1.15 (1.13–1.17) | 1.13 (1.1–1.17) | 1.63 (1.55–1.72) | 1.56 (1.49–1.64) | 1.14 (1.09–1.2) |
| **1995–1999** | 1.03 (0.998–1.07) | 1.16 (1.14–1.18) | 1.17 (1.12–1.21) | 1.12 (1.1–1.14) | 1.6 (1.55–1.66) | 1.51 (1.47–1.55) | 1.14 (1.09–1.19) |
| **2000–2004** | 1.03 (1.002–1.07) | 1.16 (1.14–1.18) | 1.18 (1.13–1.22) | 1.15 (1.11–1.2) | 1.71 (1.6–1.82) | 1.58 (1.49–1.68) | 1.15 (1.08–1.21) |
| **2005–2009** | 1.04 (1.002–1.07) | 1.16 (1.14–1.18) | 1.17 (1.13–1.22) | 1.15 (1.1–1.2) | 1.7 (1.58–1.82) | 1.58 (1.48–1.68) | 1.16 (1.13–1.2) |
| **2010–2014** | 1.03 (1–1.07) | 1.16 (1.14–1.18) | 1.18 (1.14–1.22) | 1.15 (1.11–1.2) | 1.7 (1.57–1.83) | 1.6 (1.5–1.71) | 1.16 (1.08–1.25) |
| **2015–2017** | 1.04 (1.01–1.07) | 1.15 (1.13–1.17) | 1.17 (1.12–1.22) | 1.15 (1.1–1.2) | 1.7 (1.59–1.81) | 1.6 (1.51–1.7) | 1.13 (1.08–1.19) |

RR = rate ratio; CI = confidence interval

caring for their own children [12], or through exposure to sick children while working in day-care centers. Adults are a potential reservoir for exposure to pertussis in very young infants [2], although the exposure should be the same regardless of the infant's sex. Such possible sex differences in exposure, vaccination rates [37] or medical services utilization are not relevant explanations for the excess pertussis incidence rates observed in infants and young children.

While this study cannot provide information on the mechanisms underlying the excess incidence rates in females, we can explore some possible explanations. Sex differences in pertussis incidence rates can be due to factors such as biological differences between sexes, such as sex chromosomes and sex hormones. It could be postulated that genetic and/or hormonal differences explain, at least partly, increased pertussis incidence rates in females. Studies indicate that infection with B. pertussis results in an immune response mediated through expansion of Th17 cells [38, 39]. These cells may induce tissue immunopathology [40] via the production of inflammatory cytokines and the creation of an environment contributing to inflammation of the upper respiratory tract, duration of lung tissue pathology and prolonged cough [40, 41].

Differences in the immune responses between males and females are in part attributed to the X chromosome, which contains a high number of immune-related genes and regulatory factors that are involved in both the innate and adaptive immune responses [42, 43]. X-linked mosaicism encourages a highly polymorphic gene expression that could enhance the immune response more in females [43], with consequent more symptomatic pertussis. Thus, a stronger immune response in females could result in more clinical manifestations of pertussis.

Sex hormones may also be implicated in the higher incidence rates of clinical pertussis in females. Higher pertussis incidence and immune response may also be due in part to an estrogen mediated enhanced pro-inflammatory response to B. pertussis invasion via IL-17 and a cytokine storm phenomenon. Progesterone and estrogen are lead to more severe inflammation in respiratory diseases [44, 45] and an increased expression of IL-17, whereas testosterone [46] reduces the generation of Th17 cells. Kuwabara T et al [47] showed that IL-17 plays an important role in chronic inflammation that occurs during the pathogenesis of autoimmune diseases such as human rheumatoid arthritis and multiple sclerosis (MS). Pertussis toxin served as adjuvants to induce sensitization to neural antigens in experimental autoimmune encephalomyelitis, the principle animal model of MS, which is more common in female. It appears that IL-17, the cytokine that is involved in pertussis pathogen eradication [38] and autoimmune

diseases pathways [47], along with associated chemokines IL-1β, IL-23R, IL-6 and many others [44, 47] is significant and may be linked to an estrogen-regulated immune overresponse to pertussis infection in female.

The impact of sex hormones on the immune response prior to puberty especially in infancy, is not clear. Maternal hormones that pass through the placenta affect male and female fetuses equally [48]. The mini-puberty phenomenon in infancy results in higher endogenous estrogen levels in female infants [49], which could explain the higher incidence of disease in female infants. In addition, maternal hormones may persist in the infant's circulation for some months after birth and will affect females and males equally [48]. This could mitigate the sex difference in disease and explain the lower female to male incidence ratios in infants than those seen at older ages.

Differences in sex hormone levels continue in childhood [50] and in pre-pubertal children [51].

It is conceivable that, in young adulthood, hormonal and genetic differences continue to exist, but the excess pertussis incidence rates need to be viewed in the context of possible different exposure. It has been noted that IL-17 blood levels increase in pregnancy [52, 53].The available literature [54] indicates that the immune response of aged women may be preserved to a greater extent than in aged men and may contribute to prolonged inflammatory responses and tissue damage in respiratory airways. This could be the reason why women exhibit a higher pertussis incidence rate even in older ages.

## Conclusions

This study has provided strong evidence that while the excess female incidence rates for pertussis observed in all age groups differ in magnitude, they are consistent over a number of countries and over different time periods. The mechanism underlying the excess in females is still largely unknown. Behavioral factors may contribute to some of the differences seen in the post-pubertal age groups. However, in infants and children, genetic factors, as well as sex hormones could play a part. Our findings suggest the need to explore further the role of sex differences in the mechanism of pertussis infection, when evaluating the efficacy of pertussis vaccine dosing and schedules especially in adult females for disease prevention and public health promotion.

## Supporting information

**S1 Data.**
(XLSX)

## Acknowledgments

We thank the official representative of Public Health England, the Israeli Ministry of Health, the official representative of RIVM, Netherlands, and to all the official institutions of all other countries for the providing their national data on pertussis incidence.

## Author Contributions

**Conceptualization:** Victoria Peer.

**Data curation:** Victoria Peer.

**Formal analysis:** Naama Schwartz.

**Investigation:** Victoria Peer.

**Methodology:** Manfred S. Green.

**Project administration:** Victoria Peer.

**Software:** Naama Schwartz.

**Supervision:** Manfred S. Green.

**Visualization:** Manfred S. Green.

**Writing – original draft:** Victoria Peer.

**Writing – review & editing:** Naama Schwartz, Manfred S. Green.

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
