## [Decision Letter · Decision Letter 0]

28 Jan 2020

PONE-D-19-34094

A multi-country, multi-year, meta-analytic evaluation of the sex differences in age-specific pertussis incidence rates

PLOS ONE

Dear Dr Victoria Peer,

Thank you for submitting your manuscript to PLOS ONE. After careful consideration, we feel that it has merit but does not fully meet PLOS ONE’s publication criteria as it currently stands. Therefore, we invite you to submit a revised version of the manuscript that addresses the points suggested by both reviewers.

We would appreciate receiving your revised manuscript by February 27. To enhance the reproducibility of your results, we recommend that if applicable you deposit your laboratory protocols in protocols.io, where a protocol can be assigned its own identifier (DOI) such that it can be cited independently in the future. For instructions see: http://journals.plos.org/plosone/s/submission-guidelines#loc-laboratory-protocols

We look forward to receiving your revised manuscript.

Kind regards,

Daniela Flavia Hozbor

Academic Editor

PLOS ONE

Journal Requirements:

2. At this time, we ask that you please provide the dates on which the original search was performed in the Methods section of your manuscript.

Reviewers' comments:

Reviewer's Responses to Questions

**Comments to the Author**

1. Is the manuscript technically sound, and do the data support the conclusions?

Reviewer #1: Yes

Reviewer #2: Yes

2. Has the statistical analysis been performed appropriately and rigorously? 

Reviewer #1: Yes

Reviewer #2: Yes

3. Have the authors made all data underlying the findings in their manuscript fully available?

Reviewer #1: Yes

Reviewer #2: No

4. Is the manuscript presented in an intelligible fashion and written in standard English?

Reviewer #1: No

Reviewer #2: Yes

5. Review Comments to the Author

Reviewer #1: Review of Peer et al PLOS ONE 2020

The authors present a meta analysis of public health surveillance data from 9 countries over a 27 year span to address a persistent question within the pertussis research community: is pertussis more common among females than males? The answer appears to be yes.

Overall, I was very pleased to see this paper and found the methodology and results to be persuasive. Overall, the paper would benefit from a final review to correct small grammatical errors in case and tense, but these were of minor concern.

I had several more substantive comments that I think could be addressed quite easily in a revision as follows:

1. I think that the background section in the abstract does not really get to the main point of the paper very clearly. Yes, sex differences could be an important clue to pathogenesis or exposure risk, but that’s not the starting point for this analysis so much as the relevance of the finding (if confirmed) on the back end. It seems odd to start with justifying the relevance of the finding before confirming that the finding is correct. I would instead start, “Numerous studies over the years have suggested pertussis is more common in females than males, but the quality of these studies and risk for selection biases, make it less clear whether a true sex difference exists. To address this question, we conducted a meta analysis of prospectively collected surveillance data from 9 countries over a 27 year span.”

2. Lines 48-49, maybe simplify sentence to state that pertussis can be mild, severe or potentially fatal, with severe and fatal infections concentrated among infants.

3. Line 66, sentence ends abruptly and without resolution, ‘Facilities and mandatory.’ I assume what you meant is that these data were generated in settings with mandatory reporting requirements for certain infectious diseases, including pertussis.

4. Incidence rates imply time. None of your results report as events/100,000 per unit time. Presumably the incidence rates are per year, but this needs to be stated throughout and in the tables. Please add that dimension to make these rates, not proportions.

5. Table 1. Data from Czech republic stand out with very low reported events. This makes me doubt the validity of the Czech data more than it makes me suspect that pertussis is really so different there than elsewhere. Please comment. Should these data be included? I am skeptical of them.

6. Figures are NOT labeled. There is no figure legend either. Please label the figures. Please provide a legend. Also, quality of figures is very poor. Please upload hi definition Tiff figures.

7. I see no value to the funnel plots. These are used to detect publication bias in the scientific literature. But your data are systematically collected, nationally representative, longitudinal infectious disease surveillance data and are not published in the usual sense when doing a meta-analysis. Personally, I think the funnel plots and Egger’s test play no useful role in this context and should be dropped as irrelevant and more likely misleading.

8. Line 233, authors assert that the 9 countries had similar socioeconomic status. That is really debatable. The Czech republic in 1990 was not comparable in wealth the UK or Canada at that time. Similarly, the data from New Zealand and Australia will be dominated by pertussis events in the aboriginal and Maori populations, who tend to be at the very bottom of wealth quintiles. Basically, I reject your assertion.

9. Not sure of the relevance of citation 9. No one is arguing that more males than females get pertussis.

10. The hormone hypothesis is intriguing and plausible. One facet of your data that supports this is the relatively small F:M gradient among infants compared with all older age groups. That is notable because estrogen levels in a baby are largely reflective of maternal estrogens, and hence tend to be more similar between boy and girl babies during early infancy. As endogenous estrogens rise, the F:M gradient increases. That seems like an important point to buttress your overall argument.

Reviewer #2: General comments:

I enjoyed reading this ms. Looking at sex specific differences is definitely important, due to the differences the authors were able to identify.

For this reason, a figure of the time series for each country by sex would have given the reader a good motivation for the study. I recommend plotting that. This would then be followed by the RR analysis.

There were a few things however that I suggest revising. Please see below.

Introduction:

Line 46 you needs to add references to the possible hypotheses, also you aren’t listing all possible hypotheses so I recommend using “for example” or “among others”

You need to italicize Bordetella pertussis

Line 47-48 do you mean manifestations of “disease”? Disease and infection are two different things.

Line 50 The reports aren’t controversial, some of the assumptions are sometimes controversial. I advise revising this sentence.

Also Skoff et al was a robust study and seminal in understanding age specific transmission, so I believe it warrants less inflammatory remarks.

Line 53- you are forgetting social contacts, which have shown to be a major driver in risk.

Line 57 Are you sure you mean mechanisms of infection or transmission dynamics?

Line 59 – what are these reliable denominators (odd way to finish the introduction) please expand.

Material and methods:

Line 64-65- what do you mean by sophisticated lab facilities?

Line 66-7- Are you making the data you gathered available?

Are these country data resolved by sex?

Line 87-89- I think you can remove this

Line 120 “used”

Results:

Line 239: you mean x axis, right?

What are the grey squares? They make it very hard to look at the variation between years.

While I understand the choice of blocks of 5 years, I wonder if you should have an analysis specific to epidemic years. For instance, in England 2012 was an outbreak year.

In the “leave one out “you should actually consider specific years rather than blocks of years.

Line 153 - you need a space after Fig 3

For table 3 – You should either use individual years or blocks of years. In some countries you have 2015 alone. This is confusing. In blocks of years you might be getting rid of between year variation, especially if you think of epidemic years. Also for instance in the England data set, after 2004 there were some change in reporting this can be a source of further confusion.

Line 196 - So the fact that age group contributes almost all the variation is not surprising. This is due to contact structure and age specific infection risk. This has been shown by Rohani et al 2010, Skoff et al 2015, Bento et al 2018.

Discussion:

Line 230- again what do you mean by “reliable denominators”

Line 237 – I don’t understand this sentence. Revise it.

Line 244- Are you assuming all individuals in your study have been vaccinated? Otherwise this sentence needs revision… Remember that individuals in older age categories (+65) were for the most part not vaccinated. Also the ones younger than 1 many of these cases are kinds younger than 2 mo.

Line 281- your point here is relevant and pertinent. What could be driving these differences prior to puberty?

Also, how about difference investments by sex in terms of reproduction vs immunity?

6. PLOS authors have the option to publish the peer review history of their article (what does this mean?). If published, this will include your full peer review and any attached files.

Reviewer #1: Yes: Christopher J Gill

Reviewer #2: No

---

## [Author Response · Author response to Decision Letter 0]

20 Feb 2020

PONE-D-19-34094

A multi-country, multi-year, meta-analytic evaluation of the sex differences in age-specific pertussis incidence rates

PLOS ONE

Journal Requirements:

Comment: When submitting your revision, we need you to address these additional requirements.

Response: We will structure the article in accordance with PLOS ONE's style requirements.

Comment: At this time, we ask that you please provide the dates on which the original search was performed in the Methods section of your manuscript.

Response: Now we provided this information in the Methods section.

Comment: PLOS requires an ORCID iD for the corresponding author in Editorial Manager on papers submitted after December 6th, 2016. Please ensure that you have an ORCID iD and that it is validated in Editorial Manager. To do this, go to ‘Update my Information’ (in the upper left-hand corner of the main menu), and click on the Fetch/Validate link next to the ORCID field. This will take you to the ORCID site and allow you to create a new iD or authenticate a pre-existing iD in Editorial Manager. Please see the following video for instructions on linking an ORCID iD to your Editorial Manager account: https://www.youtube.com/watch?v=_xcclfuvtxQ

Response : I created the new ORCID iD and validated in Editorial Manager.

Comment: Please include a separate caption for each figure in your manuscript.

Response: Done. Captions for each figure are part of RESULTS section. 

Response to reviewers

Many thanks to the reviewers for their important and constructive reviews. We have attempted to address all their comments and have corrected the manuscript accordingly. Below are the details of our responses to the reviewers’ comments

Reviewers' comments:

Reviewer's Responses to Questions

Reviewer 1

Comments to the Author

1. Is the manuscript technically sound, and do the data support the conclusions?

Reviewer #1: Yes

Reviewer #2: Yes

 2. Has the statistical analysis been performed appropriately and rigorously? 

Reviewer #1: Yes

Reviewer #2: Yes

3. Comment: Have the authors made all data underlying the findings in their manuscript fully available?

Reviewer #1: Yes

Reviewer #2: No

Response: all raw data underlying the findings described in the manuscript fully available in the table number 1. Due to the large amount of data for presentation purposes group of years together were used. 

If needed, we will make available all the data by each calendar year for each country separately and upload as a supplementary table (as a part of The PLOS Data policy requirement).

4. Is the manuscript presented in an intelligible fashion and written in standard English?

Reviewer #1: No

Response: Done. Grammatical errors are corrected.

Reviewer #2: Yes

5. Review Comments to the Author

Reviewer #1: Review of Peer et al PLOS ONE 2020

The authors present a meta analysis of public health surveillance data from 9 countries over a 27 year span to address a persistent question within the pertussis research community: is pertussis more common among females than males? The answer appears to be yes.Overall, I was very pleased to see this paper and found the methodology and results to be persuasive. Overall, the paper would benefit from a final review to correct small grammatical errors in case and tense, but these were of minor concern.

I had several more substantive comments that I think could be addressed quite easily in a revision as follows:

Comment 1:

I think that the background section in the abstract does not really get to the main point of the paper very clearly. Yes, sex differences could be an important clue to pathogenesis or exposure risk, but that’s not the starting point for this analysis so much as the relevance of the finding (if confirmed) on the back end. It seems odd to start with justifying the relevance of the finding before confirming that the finding is correct. I would instead start, “Numerous studies over the years have suggested pertussis is more common in females than males, but the quality of these studies and risk for selection biases, make it less clear whether a true sex difference exists. To address this question, we conducted a meta analysis of prospectively collected surveillance data from 9 countries over a 27 year span.”

Response: this part of abstract revised again and rewritten.

Comment 2:

Lines 48-49, maybe simplify sentence to state that pertussis can be mild, severe or potentially fatal, with severe and fatal infections concentrated among infants.

Response: this sentence is simplified and rewritten.

Comment 3:

 Line 66, sentence ends abruptly and without resolution, ‘Facilities and mandatory.’ I assume what you meant is that these data were generated in settings with mandatory reporting requirements for certain infectious diseases, including pertussis.

Response: this sentence is rewritten 

Comment 4:

Incidence rates imply time. None of your results report as events/100,000 per unit time. Presumably the incidence rates are per year, but this needs to be stated throughout and in the tables. Please add that dimension to make these rates, not proportions.

Response: incidence rates produced are annual incidence rates(We stated it in the Statistical Analyses section)

Comment 5:

Table 1. Data from Czech republic stand out with very low reported events. This makes me doubt the validity of the Czech data more than it makes me suspect that pertussis is really so different there than elsewhere. Please comment. Should these data be included? I am skeptical of them.

Response: We agree with you and understand the problem. We carried out the sensitivity analysis to determine whether the effects we observed were affected by one specific country and did not find this to be so. We believe that very low reported events in Czech Republic are not a source of selection bias and the low rates are unlikely to be different for females and males.

Comment 6:

Figures are NOT labeled. There is no figure legend either. Please label the figures. Please provide a legend. Also, quality of figures is very poor. Please upload hi definition Tiff figures.

Response: All figures are labeled and figures legend will be provided. All labeled figures will be uploaded as TIFF files.

Comment 7:

I see no value to the funnel plots. These are used to detect publication bias in the scientific literature. But your data are systematically collected, nationally representative, longitudinal infectious disease surveillance data and are not published in the usual sense when doing a meta-analysis. Personally, I think the funnel plots and Egger’s test play no useful role in this context and should be dropped as irrelevant and more likely misleading.

Response: funnel plots provide a further examination of whether there are outlier countries related to the size of the incidence RR

Comment 8:

Line 233, authors assert that the 9 countries had similar socioeconomic status. That is really debatable. The Czech republic in 1990 was not comparable in wealth the UK or Canada at that time. Similarly, the data from New Zealand and Australia will be dominated by pertussis events in the aboriginal and Maori populations, who tend to be at the very bottom of wealth quintiles. Basically, I reject your assertion.

Response: We totally accept your comment. We mean that we included countries with advanced health system and facilities. This sentence is rewritten

Comment 9:

Not sure of the relevance of citation 9. No one is arguing that more males than females get pertussis.

Response: We think it is still important to show that there have been publications suggesting the opposite. In fact, in most studies of other infectious diseases, males have higher morbidity. In our study on viral meningitis we showed that the higher incidence rates from viral meningitis in males under the age of 15 is remarkably consistent across countries and time-periods (Peer V, Schwartz N, Green MS. Consistent, Excess Viral Meningitis Incidence Rates in Young Males: A Multi-country, Multi-year, Meta-analysis of National Data. The Importance of Sex as a Biological Variable.EClinicalMedicine. 2019;15:62-71)

Comment 10:

The hormone hypothesis is intriguing and plausible. One facet of your data that supports this is the relatively small F:M gradient among infants compared with all older age groups. That is notable because estrogen levels in a baby are largely reflective of maternal estrogens, and hence tend to be more similar between boy and girl babies during early infancy. As endogenous estrogens rise, the F:M gradient increases. That seems like an important point to buttress your overall argument.

Response: Hormones from the mother pass through the placenta into the baby's blood during pregnancy. Pregnant women produce high levels of the hormone estrogen that can affect the baby.

By the second week after birth, hormones are no longer present in the infant. 

(Gevers EF, Fischer DA, Dattani MT. Fetal and neonatal endocrinology. In: Jameson JL, De Groot LJ, de Kretser DM, et al, eds. Endocrinology: Adult and Pediatric. 7th ed. Philadelphia, PA: Elsevier Saunders; 2016:chap 145)

Reviewer #2: General comments:

Comment 1:

I enjoyed reading this ms. Looking at sex specific differences is definitely important, due to the differences the authors were able to identify. For this reason, a figure of the time series for each country by sex would have given the reader a good motivation for the study. I recommend plotting that. This would then be followed by the RR analysis.

Response:We considered the option of a time series analysis, but after consultation felt it would not contribute to the main issue of the paper, since there were no time-related effects.

There were a few things however that I suggest revising. Please see below.

Comment 2:

Introduction:

Line 46 you needs to add references to the possible hypotheses, also you aren’t listing all possible hypotheses so I recommend using “for example” or “among others”

Response: We rewrote this sentence and added the relevant references. 

Comment 3:

You need to italicize Bordetella pertussis

Response: Done

Comment 4:

Line 47-48 do you mean manifestations of “disease”? Disease and infection are two different things.

Response: We mean ''disease''. This sentence is rewritten.

Comment 5:

Line 50 The reports aren’t controversial, some of the assumptions are sometimes controversial. I advise revising this sentence.

Response: This sentence is rewritten.

Comment 6:

Also Skoff et al was a robust study and seminal in understanding age specific transmission, so I believe it warrants less inflammatory remarks.

Response: We mentioned Skoff et al as robust study in our manuscript.

Comment 7:

Line 53- you are forgetting social contacts, which have shown to be a major driver in risk.

Response: This sentence is rewritten and reference is added

Comment 8:

Line 57 Are you sure you mean mechanisms of infection or transmission dynamics?

Response: This sentence is rewritten 

Comment 9:

Line 59 – what are these reliable denominators (odd way to finish the introduction) please expand.

Response: This sentence is rewritten

Comment 10:

Material and methods:

Line 64-65- what do you mean by sophisticated lab facilities?

Response: This sentence is rewritten

Comment 11:

Line 66-7- Are you making the data you gathered available?

Response: Yes, all the data is available

Comment 12:

Are these country data resolved by sex?

Response: Yes, all the data in these countries is resolved by sex

Comment 13:

Line 87-89- I think you can remove this

Response: We think it's important to clarify the issue of Ethical considerations

Comment 14:

Line 120 “used”

Response: Done

Comment 15:

Results: Line 239: you mean x axis, right?

Response: This error is corrected.

Comment 16:

What are the grey squares? They make it very hard to look at the variation between years.

Response: This is the software default

Comment 17:

While I understand the choice of blocks of 5 years, I wonder if you should have an analysis specific to epidemic years. For instance, in England 2012 was an outbreak year. In the “leave one out” “ you should actually consider specific years rather than blocks of years.

Response: To evaluate the effect of this particular year on the incidence of pertussis, we performed leave-one-out sensitivity analysis by single year and recomputed the pooled RRs. The pooled RRs calculated after leave-one-out sensitivity analysis by single year didn't add to the strong results. We don’t have any reason to believe that the results would differ for years with higher incidence rates.

Comment 18:

Line 153 - you need a space after Fig 3

Response: Done

Comment 19:

For table 3 – You should either use individual years or blocks of years. In some countries you have 2015 alone. This is confusing. In blocks of years you might be getting rid of between year variation, especially if you think of epidemic years.

Response: We agree that some countries database differ in reported years, but have a common period.

In the beginning of results processing we performed the meta-analysis by single year (for every age group and country).it's impossible to perform them in the manuscript as a plot because of huge amount of the data). Year's series allowed us to perform the meta-analysis (forest plots) in appropriate way. Of course, if there are only 2015 data, then the result applies to this particular year.

Table 3 displays the pooled RR per age group for all of the reported years and countries and for the common period as well.

To evaluate the effect of reported year on the incidence of pertussis, we performed leave-one-out sensitivity analysis by single year and recomputed the pooled RRs. The estimated pooled RRs calculated after leave-one-out sensitivity analysis by single year didn't show differences from the primary values of time series, thus no single country or period of time affected the pooled RRs. 

Comment 20: 

Also for instance in the England data set, after 2004 there were some change in reporting this can be a source of further confusion.

Response: We think that, even the reporting system has changed in England it has not created a selection bias, which would affect differently males and females. 

Comment 21:

Line 196 - So the fact that age group contributes almost all the variation is not surprising. This is due to contact structure and age specific infection risk. This has been shown by Rohani et al 2010, Skoff et al 2015, Bento et al 2018.

Response: there are references in the manuscript that address this point.

1. Skoff TH, Hadler S, Hariri S. The epidemiology of nationally reported pertussis in the United States, 2000–2016. Clin Infect Dis. 2019; 68: 1634– 40.

2. Wensley A, Hughes GJ, Campbell H, et al. Risk factors for pertussis in adults and teenagers in England. Epidemiol Infect. 2017;145:1025–36.

Comment 22:

Discussion:

Line 230- again what do you mean by “reliable denominators”

Response: This sentence is rewritten

Comment 23:

Line 237 – I don’t understand this sentence. Revise it.

Response: This sentence is rewritten

Comment 24:

Line 244- Are you assuming all individuals in your study have been vaccinated? Otherwise this sentence needs revision… Remember that individuals in older age categories (+65) were for the most part not vaccinated. Also the ones younger than 1 many of these cases are kinds younger than 2 mo.

Response: This sentence is rewritten.We assume that only the part of population, especially infants and children, is vaccinated. The majority of adults in older ages are not vaccinated. 

Comment 25:

Line 281- your point here is relevant and pertinent. What could be driving these differences prior to puberty?

Response: This part is rewritten. 

Markedly higher concentrations of estrogen were measured in prepubertal girls as compared to prepubertal boys in whom most values were below the detection limit. Thus, even before any physical signs of pubertal maturation, girls had significantly higher estrogen concentrations compared to boys. (Frederiksen H, Johannsen TH, Andersen SE, Albrethsen J, Landersoe SK, Petersen JH, Andersen AN, Vestergaard ET, Schorring ME, Linneberg A, Main KM, Andersson AM, Juul A. Sex-specific estrogen levels and reference intervals from infancy to late adulthood determined by LC-MS/MS.J Clin Endocrinol Metab. 2019 )

Comment 26:

Also, how about difference investments by sex in terms of reproduction vs immunity?

Response: We addressed the issue of pregnancy in the final part of the discussion. We assume that 5 -10 % of women in the age group 15-44 are pregnant.

Placental immune response for specific viruses and pathogens affect the pregnant woman’s susceptibility to and severity of certain infectious diseases. The generalization of pregnancy as a condition of general immune suppression or increased risk is misleading.( Mor G, Cardenas I. The immune system in pregnancy: a unique complexity. Am J Reprod Immunol. 2010; 63:425-33)

There is growing evidence that the type of response initiated by the placenta might determine the immunologic response of the mother . It is now clear, the placenta represents important immune modulator that affect the global response of the mother to microbial infections. (Racicot K, Kwon JY, Aldo P, Silasi M, Mor G. Understanding the complexity of the immune system during pregnancy. Am J Reprod Immunol. 2014 ;72: 107-16).

Additional amendments from 20-02-2020:

PONE-D-19-34094R1

A multi-country, multi-year, meta-analytic evaluation of the sex differences in age-specific pertussis incidence rates

Thank you for submitting your manuscript entitled "A multi-country, multi-year, meta-analytic evaluation of the sex differences in age-specific pertussis incidence rates" to PLOS ONE. Your manuscript files have been checked in-house but before we can proceed we need you to address the following issues:

1) Please ensure that you refer to Figure 8 in your text as, if accepted, production will need this reference to link the reader to the figure.

Answer: We referred to Figure 8 in the text

2) Please amend your list of authors on the manuscript to ensure that each author is linked to an affiliation.

We note that you have included affiliation numbers 1,¶ and * however only affiliations 1 has authors linked to them. Please amend affiliation 4 to link an author to it or remove if added in error.

Answer: We amended the list of authors and affiliation on the title page

3) Please allow me to provide more insight into how much of your data you are meant to publicly share and what exactly constitutes a minimal data set or underlying data.

We require authors to share the “minimal data set” for their submission. PLOS defines the minimal data set to consist of the data required to replicate all study findings reported in the article, as well as related metadata and methods. Additionally, PLOS requires that authors comply with field-specific standards for preparation, recording, and deposition of data when applicable. For example, authors should submit the following data:

3) The points extracted from images for analysis.

Authors do not need to submit their entire data set if only a portion of the data were used in the reported study. Also, authors do not need to submit the raw data collected during an investigation if the standard in the field is to share data that have been processed.

Answer: All relevant data are within the manuscript.

Table number 1 contains all the minimal data used for the study ( for meta-analysis).

On this table we performed all the data (“minimal data set” for all reported years together) about pertussis cases by sex (cumulative total n), total population by sex (cumulative total N), calculated incidence rate per 100 000 male or female population, and incidence rate ratio (female: male incidence rate ratio).All this data required for study findings replication. 

 If needed we could share the all raw data by single year and sex for each country (pertussis cases by sex n by single year and total population by sex N by single year).

Additionally, we require authors to provide sample image data in support of all reported results (e.g. for immunohistochemistry images, fMRI images, etc.), either with the submission files or in a public repository.

Answer: We have no any other image data in support of our findings

At this time, please confirm that the minimal data set are within your paper.

Answer: We confirm that the minimal data set are within the paper.

---

## [Decision Letter · Decision Letter 1]

16 Mar 2020

PONE-D-19-34094R1

A multi-country, multi-year, meta-analytic evaluation of the sex differences in age-specific pertussis incidence rates

PLOS ONE

Dear Dr Victoria Peer,

Thank you for submitting your manuscript to PLOS ONE. After careful consideration, we feel that it has merit but does not fully meet PLOS ONE’s publication criteria as it currently stands. Therefore, we invite you to submit a revised version of the manuscript that addresses the following minor point to ensure clarity of reporting. In the manuscript text, the authors frequently describe their study using the term 'meta-analysis', we suggest to replace this term by 'meta-analytic methods' to describe their analyses.

We would appreciate receiving your revised manuscript by April 1. To enhance the reproducibility of your results, we recommend that if applicable you deposit your laboratory protocols in protocols.io, where a protocol can be assigned its own identifier (DOI) such that it can be cited independently in the future. For instructions see: http://journals.plos.org/plosone/s/submission-guidelines#loc-laboratory-protocols

We look forward to receiving your revised manuscript.

Kind regards,

Daniela Flavia Hozbor

Academic Editor

PLOS ONE

Reviewers' comments:

Reviewer's Responses to Questions

**Comments to the Author**

1. If the authors have adequately addressed your comments raised in a previous round of review and you feel that this manuscript is now acceptable for publication, you may indicate that here to bypass the “Comments to the Author” section, enter your conflict of interest statement in the “Confidential to Editor” section, and submit your "Accept" recommendation.

Reviewer #1: All comments have been addressed

2. Is the manuscript technically sound, and do the data support the conclusions?

Reviewer #1: Yes

3. Has the statistical analysis been performed appropriately and rigorously? 

Reviewer #1: Yes

4. Have the authors made all data underlying the findings in their manuscript fully available?

Reviewer #1: Yes

5. Is the manuscript presented in an intelligible fashion and written in standard English?

Reviewer #1: Yes

6. Review Comments to the Author

Reviewer #1: Re. comment 10. While it is true that maternal estrogens disappear from fetal circulation within a few weeks of birth the biological effects of the in utero estrogen exposure will linger for months. Hence my comment about the reduced M:F gradient among infants still stands as a plausible explanation. And as I stated before, I believe that this strengthens your argument.

7. PLOS authors have the option to publish the peer review history of their article (what does this mean?). If published, this will include your full peer review and any attached files.

Reviewer #1: Yes: Christopher Gill

---

## [Author Response · Author response to Decision Letter 1]

22 Mar 2020

PONE-D-19-34094R1

A multi-country, multi-year, meta-analytic evaluation of the sex differences in age-specific pertussis incidence rates

PLOS ONE

Dear Dr Victoria Peer,

Thank you for submitting your manuscript to PLOS ONE. After careful consideration, we feel that it has merit but does not fully meet PLOS ONE’s publication criteria as it currently stands. Therefore, we invite you to submit a revised version of the manuscript that addresses the following minor point to ensure clarity of reporting. In the manuscript text, the authors frequently describe their study using the term 'meta-analysis', we suggest to replace this term by ' meta-analytic methods ' to describe their analyses.

Response:

Dear Dr.Daniela Flavia Hozbor,

We replaced the term 'meta-analysis' with term 'meta-analytic methods' throughout the article.

Reviewers' comments:

Reviewer's Responses to Questions

Comments to the Author

1. If the authors have adequately addressed your comments raised in a previous round of review and you feel that this manuscript is now acceptable for publication, you may indicate that here to bypass the “Comments to the Author” section, enter your conflict of interest statement in the “Confidential to Editor” section, and submit your "Accept" recommendation.

Reviewer #1: All comments have been addressed

2. Is the manuscript technically sound, and do the data support the conclusions?

Reviewer #1: Yes

3. Has the statistical analysis been performed appropriately and rigorously? 

Reviewer #1: Yes

4. Have the authors made all data underlying the findings in their manuscript fully available?

Reviewer #1: Yes

5. Is the manuscript presented in an intelligible fashion and written in standard English?

Reviewer #1: Yes

6. Review Comments to the Author

Reviewer #1: Re. comment 10. While it is true that maternal estrogens disappear from fetal circulation within a few weeks of birth the biological effects of the in utero estrogen exposure will linger for months. Hence my comment about the reduced M:F gradient among infants still stands as a plausible explanation. And as I stated before, I believe that this strengthens your argument.

Response: We completely agree with the reviewer and think that the reviewer's comment may be another possible mechanism and explanation for the results. We have now added that to the discussion on lines 289-295 in the revised manuscript. 

7. PLOS authors have the option to publish the peer review history of their article (what does this mean?). If published, this will include your full peer review and any attached files.

Do you want your identity to be public for this peer review? For information about this choice, including consent withdrawal, please see our Privacy Policy.

Reviewer #1: Yes: Christopher Gill

---

## [Editor Report · Decision Letter 2]

27 Mar 2020

A multi-country, multi-year, meta-analytic evaluation of the sex differences in age-specific pertussis incidence rates

PONE-D-19-34094R2

Dear Dr. Victoria Peer,

We are pleased to inform you that your manuscript has been judged scientifically suitable for publication and will be formally accepted for publication once it complies with all outstanding technical requirements.

With kind regards,

Daniela Flavia Hozbor

Academic Editor

PLOS ONE
---

## [Editor Report · Acceptance letter]

10 Apr 2020

PONE-D-19-34094R2 

A multi-country, multi-year, meta-analytic evaluation of the sex differences in age-specific pertussis incidence rates 

Dear Dr. Peer:

I am pleased to inform you that your manuscript has been deemed suitable for publication in PLOS ONE. Congratulations! Your manuscript is now with our production department. 

With kind regards,

on behalf of

Dr. Daniela Flavia Hozbor 

Academic Editor

PLOS ONE